# Investigating the Potential of Isolating and Expanding Tumour-Infiltrating Lymphocytes from Adult Sarcoma

**DOI:** 10.3390/cancers14030548

**Published:** 2022-01-21

**Authors:** Alice Ko, Victoria S. Coward, Nalan Gokgoz, Brendan C. Dickson, Kim Tsoi, Jay S. Wunder, Irene L. Andrulis

**Affiliations:** 1Department of Laboratory Medicine & Pathobiology, University of Toronto, Toronto, ON M5S 1A8, Canada; brendan.dickson@sinaihealth.ca (B.C.D.); andrulis@lunenfeld.ca (I.L.A.); 2Department of Molecular Genetics, University of Toronto, Toronto, ON M5S 1A8, Canada; vcoward@lunenfeld.ca; 3Lunenfeld-Tanenbaum Research Institute, Sinai Health System, Toronto, ON M5G 1X5, Canada; nalan@lunenfeld.ca (N.G.); jay.wunder@sinaihealth.ca (J.S.W.); 4University of Toronto Musculoskeletal Oncology Unit, Sinai Health System, Toronto, ON M5G 1X5, Canada; kim.tsoi@sinaihealth.ca; 5Department of Pathology and Laboratory Medicine, Sinai Health System, Toronto, ON M5G 1X5, Canada; 6Department of Surgery, University of Toronto, Toronto, ON M5T 1P5, Canada

**Keywords:** sarcoma, tumour-infiltrating lymphocytes, immunotherapy

## Abstract

**Simple Summary:**

Sarcomas are rare cancers that arise from connective tissue. There are more than 50 subtypes, many of which are associated with a high risk of metastasis and poor prognosis. Subtype-specific treatment is limited and conventional treatment for advanced disease has varying effects across individuals and tumour subtypes. Adoptive cell therapy shows potential to provide more personalized treatment; this study aims to explore the potential of using tumour-infiltrating lymphocytes (TIL) to treat sarcoma. We optimized a sarcoma-specific expansion protocol and successfully expanded TILs from 54 of 92 sarcoma specimens. We characterized primarily CD4^+^ and CD8^+^ T-cells in the expanded TIL cultures and demonstrated their reactivity to general stimuli. Although sarcomas in general do not have abundant lymphocytic infiltration, our expansion protocol allowed for successful expansions of viable and reactive lymphocytes, thus showing the prospects of adopting TIL therapy in sarcoma.

**Abstract:**

Sarcomas are a heterogeneous group of mesenchymal neoplasms, many of which are associated with a high risk of metastasis and poor prognosis. Conventional chemotherapy and targeted therapies have varying effects across individuals and tumour subtypes. The current therapies frequently provide limited clinical benefit; hence, more effective treatments are urgently needed. Recent advances in immunotherapy, such as checkpoint inhibition or adoptive cell therapy (ACT), show potential in increasing efficacy by providing a more personalized treatment. Therapy with tumour-infiltrating lymphocytes (TILs) is an emerging field in immunotherapy. Here, we collected 190 sarcoma tumour specimens from patients without pre-operative adjuvant treatment in order to isolate TILs. We compared different methods of TIL expansion and optimized a protocol specifically for efficacy in culturing TILs from sarcoma. The expanded TIL populations were characterized by flow cytometry analysis using CD3, CD4, CD8, CD14, CD19 and CD56 markers. The TIL populations were non-specifically stimulated to establish TIL reactivity. Through an optimized expansion protocol, TILs were isolated and cultured from 54 of 92 primary sarcoma specimens. The isolated TILs varied in CD4^+^ and CD8^+^ T-cell compositions and retained their ability to release IFNγ upon stimulation. Our results suggest that certain sarcoma subtypes have the potential to yield a sufficient number of TILs for TIL therapy.

## 1. Introduction

Among annual new malignancies in North America, sarcomas comprise approximately 1% of cases [1,2]. These neoplasms are of mesenchymal origin and ostensibly occur anywhere in the body, including bone and soft tissues. Prognosis for sarcoma patients is based on several factors: stage of disease, histopathological subtype, tumour grade, tumour size, and primary tumour location [3,4]. Furthermore, the rarity of these neoplasms is potentiated by the fact that there are over 50 morphologically and genetically distinct types and subtypes. The rarity of sarcomas renders it challenging to recruit study subjects, consequently, studies that include sarcomas amongst other malignancies often group the results of histologically and molecularly different subtypes together [5].

Depending on the sarcoma subtype, conventional treatments typically involve surgical resection, with or without consideration of (neo)adjuvant chemotherapy or radiotherapy. These treatments have differing results across the various subtypes, and often only provide short-term disease control rather than disease remission, with some subtypes exhibiting chemoresistance [6]. The overall survival rates have remained relatively stagnant at 50–60% for decades, and despite good control of localized disease, approximately 35–50% of patients experience disease reoccurrence or develop metastasis [7,8,9]. In the case of metastatic disease, the overall median survival is 8–12 months [10]. For patients with unresectable tumours or metastatic disease, aside from chemosensitive subtypes, such as Ewings sarcoma and rhabdomyosarcoma, there is limited support for the efficacy of chemotherapy on improving outcomes and radiation therapy remains exclusive to selected cases [8,10,11]. In these patients with advanced or refractory disease, treatment is mainly palliative, and focuses on reducing symptoms and improving patients’ wellbeing [12].

Given the heterogeneity of sarcomas and overall lack of therapeutic targets, developing tumour type-specific treatments remains challenging; indeed, many different tumour types are lumped into one of a few existing treatment protocols [5,13]. With limited studies dedicated towards sarcomas in general, and in particular across the various subtypes, and considering their generally poor prognosis, more robust and effective treatments are urgently needed—in this regard, immunotherapy strategies offer potential value [14].

In the past decade, immunotherapy has become a frequent option to treat patients with many late-stage cancers (e.g., gastric cancer, breast cancer, and lung cancer), as it may be less invasive and intensive compared to traditional chemotherapy or radiotherapy [15,16,17,18]. Interestingly, there are certain characteristics of a subset of sarcoma that, in theory, render them more susceptible and more likely to respond to immunotherapy. For example, some cases may harbor moderate frequencies of genetic mutations, albeit less than those found in most carcinomas, that could deliver a higher likelihood of response to immunotherapy [19,20,21]. We previously performed immunohistochemistry (IHC) staining of lymphocyte markers and immune checkpoint proteins programmed death-ligand 1 (PD-L1) and programmed cell death protein 1 (PD-1) in several sarcoma subtypes and found that a subset of undifferentiated pleomorphic sarcoma (UPS) and myxofibrosarcoma (MFS) contained tumour infiltrating lymphocytes (TILs) as well as PD-L1 and PD-1 expression [22]. Moreover, TILs positive for CD3, CD4, CD8 and PD-1 were found to be positively correlated with PD-L1 expression in the tumour, and the PD-L1 levels observed in IHC were positively correlated to quantitative reverse transcription PCR (RT-qPCR) values. This is particularly promising for sarcomas, which have been described generally as non-immunogenic [6]. In a multi-centre phase 2 study on pembrolizumab, an anti-PD-1 antibody, in patients with advanced sarcomas, there were objective responses in UPS and liposarcoma (LPS) [6]. Interestingly, in UPS cases, responses were seen in both PD-L1^+^ and PD-L1^−^ tumours [6]. The results in bone sarcomas were limited; however, partial response was seen in one of 22 osteosarcoma (OS) cases and one of five chondrosarcoma cases [6]. These results suggest that there may be individuals with specific sarcoma subtypes who might be possible candidates for adoptive cell therapy (ACT) based on their tumour characteristics.

For conventionally “non-immunogenic” tumours, such as sarcomas, immunotherapy can be promising as it capitalizes on the immune system’s ability to recognize unique neoantigens, particularly by utilizing TILs for ACT. ACT relies on the transfer of autologous or allogeneic tumour-specific T-cells that can facilitate the killing of tumour cells. TILs extracted from tumour fragments have been found to specifically lyse autologous melanoma tumour cells but not major histocompatibility complex (MHC) compatible allogeneic melanoma tumour cells or normal autologous cells [23,24,25,26]. This indicates that TILs had harbored both patient and disease specificity organically and therefore have immense potential as a treatment option. TILs derived from melanoma were also found to have specificity against multiple known and unknown antigens, suggesting that TILs may be a viable treatment option for phenotypically heterogeneous malignancies, such as sarcoma [27,28]. When compared to studies on other ACTs, adoptive transfer of TILs demonstrated less toxicity and lower severity in adverse effects [23,29,30]

The standard ACT treatment in clinical trials have been reported to require ~0.2–2 × 10^11^ tumour-specific T-cells depending on the study design [23,31,32]. The Rosenberg group has been a pioneer in TIL therapy, and their widely utilized protocol includes pre-treatment myeloablative chemotherapy starting seven days prior to TIL infusion [33]. Following TIL infusion, patients are given intravenous (IV) infusions or subcutaneous injections of high dose IL2 [33,34]. Most studies and expansion protocols target carcinomas such as melanoma, breast, and lung cancer. There have been limited data on the potential application of TILs in sarcomas and a lack of published reports of isolating sarcoma-specific TILs. Thus, a more robust method of cultivating TILs from sarcomas is urgently needed to further explore the potential of ACT for sarcoma patients and advance the use of TILs for sarcoma treatment.

## 2. Materials and Methods

### 2.1. Patient Cohort

The cohort consisted of 87 patients diagnosed with various subtypes of soft tissue sarcoma (STS) and OS, who were enrolled in the Biospecimen Repository at Sinai Health System, Toronto from October 2015 to September 2019 (Table 1). The age of the patients at the time of surgery ranged from 18 to 91 years old. From these 87 patients, 92 tumour specimens were collected and used for this study.

### 2.2. Tissue Specimens

Fresh tumour specimens were collected from patients undergoing open biopsy or surgical resection without neoadjuvant therapy for sarcoma. Surgical operations followed standard-of-care surgical procedures. All patients provided written informed consent under approved institutional ethics board (REB) protocols (Mount Sinai Hospital REB 01-0138-U). Studies were conducted according to the Declaration of Helsinki. Tissue was obtained fresh and immediately transported to the laboratory in sterile HBSS (GIBCO, Waltham, MA, USA) for processing. Tissue specimens are kept flash-frozen in liquid nitrogen until further use.

### 2.3. Quantification of PD-L1

RNA extraction was performed to quantify PD-L1 levels in the primary tumour. The extraction followed the same methods outlined in Wunder et al. [22]. A portion of each frozen tumour specimen was pulverized, re-suspended in lysis solution, and subjected to RNA extraction. cDNA was then made through the extracted RNA samples. RT-qPCR of cDNA was performed on 66 tumour samples collected from 65 patients in an Applied Biosystems 7900HT Sequence Detection System (Applied Biosystems, Waltham, MA, USA). 200 ng of total RNA was reverse transcribed, and cDNA added to Power SYBR Green PCR Master Mix (Applied Biosystems, Waltham, MA, USA), forward and reverse primers (Table 2) for both *PD-L1* (30 µM) and the control gene, *STAM2* (30 µM). Pooled cancer cell line cDNA was used to construct standard curves for *PD-L1* and *STAM2*. *PD-L1* expression was quantified as a ratio relative to *STAM2*. *PD-L1* expressions of the specimens are shown in Figure A1 of the Appendix A.

### 2.4. Immunohistochemistry

Five representative cases (2 MFS and 3 UPS) underwent IHC staining for: CD3 (clone: F7.2.38; DAKO, Agilent Technologies Santa Clara, CA, USA), CD4 (clone: SP35; Abcam, Cambridge, United Kingdom), CD8 (clone: 144B; DAKO), CD20 (clone: L26; DAKO), CD68 (clone: PG-M1; DAKO), DC-LAMP (clone: 1010E1.01; Novus Biologicals, Littleton, CO, USA), and PD-L1 (clone: SP263; VENTANA, ROCHE, Basel, Switzerland). For each case, a sarcoma pathologist (B.C.D.) provided a score from five representative high-power fields (HPFs) (40× magnification; field diameter = 0.55 mm) and the values were averaged. TIL staining was scored semi-quantitatively using a 5-tiered scale: 0 (no lymphocytes); 1 (1–10 per HPF); 2 (11–50 per HPF); 3 (51–100 per HPF); 4 (>100 per HPF); 5 (>200 per HPF). Attention was given to avoid quantifying lymphocytic aggregates/follicles; areas of necrosis; tumour periphery/capsule; and CD4^+^ histiocytes. Staining only quantified for cells where nuclear and cytoplasmic morphology were present and concordant (i.e., cytoplasmic processes, in the absence of nuclei, were not quantified). Distribution of IHC scoring is shown in Figure A2 of the Appendix B. 

### 2.5. Media

All cell cultures were maintained in complete media (CM), which was made with the following supplements added to IMDM (Lonza, Basel, Switzerland): 10% human serum AB (Gemini Bio-Products, West Sacramento, CA, USA), 25 mM HEPES (Lonza), 100 U/mL penicillin (Sigma-Aldrich, St. Louis, MO, USA), 100 mg/mL streptomycin (Sigma-Aldrich), 10 mg/mL gentamicin sulfate (Lonza), 2 mM L-glutamine (Lonza), 5.561025 M 2-mercaptoethanol (Sigma-Aldrich), 6000 IU/mL human recombinant IL-2 (Novartis, Basel, Switzerland). Media was stored and aliquoted as described by Nguyen et al. [35]. Cryopreservation medium was 10% DMSO (Sigma-Aldrich) and 90% human serum AB (Gemini Bio-Products).

### 2.6. TIL Culturing

The TIL culturing method from tumour fragments (Tumour Fragment Method (TFM)) was adopted and modified from the methods described in Nguyen et al. [35]. Flash-frozen tissue specimens were thawed, washed in HBSS, and processed into ~1mm^3^ sized tumour fragments.

Tumour fragments were then distributed into separate wells of a 24-well plate. Each well contained one tumour fragment immersed in 2 mL of complete media (CM; Table 3). Plates were incubated at 37 °C with 5% CO_2_. After 1 week, wells were pooled together based on growth density and cultured as the “high-density” culture or the “low-density” culture. Under a phase-contrast microscope at 10× objective, a ‘high-density’ well would display distinct TIL clusters, a ‘low-density’ well would display scattered TIL growth, and a ‘no growth’ well would display minimal to no TIL growth (Figure 1). Tumour fragments of the corresponding wells were collected and plated in a separate well. For three times per week, 1 mL of media from each well was replaced with fresh CM. Wells were maintained at a concentration of 0.5–2 × 10^6^ cells/mL and split when necessary. After 3 weeks of culture, parent and daughter wells were combined and re-suspended in cryopreservation media (10% DMSO (Sigma-Aldrich); 90% human serum AB (Gemini Bio-Products) for storage at −180 °C in liquid nitrogen (Medigas, Milton, ON, Canada) at the Biospecimen Repository and Processing Lab. Prior to storage in liquid nitrogen, cells were stored in a Mr. Frosty™ Freezing Container (Thermo Fisher Scientific, Waltham, MA, USA) in −80 °C for 1–7 days. Different density cultures were collected as separate samples.

### 2.7. Survival Analysis

The statistical analysis was performed using GraphPad Prism (GraphPad Software; San Diego, CA, USA) [36]. Survival was assessed as the time between the date of surgical resection and the date of death of any cause (overall survival) or date of metastasis or local recurrence (disease-free survival) for patients that presented with localized disease only. For patients with multiple samples, the TIL result for the initial sample was utilized. Outcomes for living patients were censored at the time of their last follow-up examination. Follow-up data were collected until February 2020. The analysis of TIL expansion success (no wells, fail or success as previously defined) associated with overall survival and disease-free survival was performed using the Kaplan–Meier method (log-rank test). Statistical significance was set at *p* < 0.05.

### 2.8. Flow Cytometry

Expanded cell cultures were thawed from −180 °C, washed with PBS (Wisent Bioproducts, Saint-Bruno, QC, Canada), and centrifuged at 1000 rpm for 5 min. Viable cell counts were determined using trypan blue dye exclusion on a haemocytometer and readjusted to a concentration of 1 × 10^6^ cells/mL in PBS. For each cell culture, a sample and a negative control (may be omitted if there is not enough cells) were prepared. In each tube, 1 mL of the cell solution was pipetted through the filter cap and the sample was stained with fixable viability dye. Tubes were incubated on ice for 30 min in the dark before washing with staining buffer (SB; 2% fetal calf serum/PBS). Cells were then re-suspended in 150 µL of SB and incubated with 5 µL human FC block (Miltenyi Biotec, Bergisch Gladbach, Germany) on ice for 5 min. An antibody cocktail was made based on titration results with the following markers: CD4 (BioLegend, San Diego, CA, USA), CD8 (BioLegend), CD3 (BioLegend), CD56 (BioLegend), and CD14/19 (BioLegend). The cocktail was added to the sample tube only, and both tubes were then incubated on ice for 30 min in the dark. Both tubes were washed with SB at 350× *g* for 5 min in centrifuge with a swinging bucket rotor (Eppendorf, Hamburg, Germany). The supernatant was decanted, and cells were re-suspended in 2–4% formaldehyde (Sigma-Aldrich)/PBS (Wisent Bioproducts) for 30 min–1 h at room temperature (RT). Following fixation, cells were washed with SB and re-suspended in 500 µL of SB. Tubes were refrigerated in the dark until analysis. Samples were also stained with LIVE/DEAD™ Fixable Blue Dead Cell Stain (Thermo Fisher Scientific) to screen out dead cells during analysis. Data were acquired on Fortessa X-20 with HTS flow cytometer (BD) (BD Biosciences, Franklin Lakes, NJ, USA). The analysis was completed using Kaluza software (Beckman Coulter, Brea, CA, USA) [37].

#### Peripheral Blood Control

A sample of commercially available peripheral blood was used as a reference population. Both positive and negative controls were prepared using the peripheral blood, where the antibody cocktail was only added to the positive control. Each tube had 100 µL of blood that was incubated on ice for 30 min in the dark before adding 2 mL BD FACS lysing solution (BD Biosciences). The tubes were then incubated for another 15 min in RT. Tubes were centrifuged at 300× *g* for 5 min in RT, and then re-suspended in 500 µL SB.

Compensation controls were prepared for each stain using UltraComp eBeads™ (Invitrogen, Waltham, MA, USA) in order to generate a compensation grid for more accurate analysis. A separate control was prepared for the reconstituted LIVE/DEAD™ Fixable Blue Dead Cell Stain (Thermo Fisher Scientific) using ArC reactive bead (Thermo Fisher Scientific). All controls were prepared by following the manufacturer’s recommended protocol.

For each sample, if possible, a fluorescence minus one (FMO) control without CD14 and CD19 was prepared. Under ideal circumstances, an additional FMO control without CD8 was prepared.

For more detailed analysis, selected populations were further characterized at Princess Margaret Cancer Centre with additional markers (CD69, HLA-DR, T-bet, CD45RA, Ki67, CD39, PD-1, and FOXP3). Data were acquired on a BD LSRFortessa 2 flow cytometer (BD Biosciences). The analysis was completed on FlowJo software (BD Life Sciences, Ashland, OR, USA) [38].

### 2.9. Functional Assay via IFNγ ELISA

Following expansion, TILs were stimulated in CM supplemented with phorbol myristate acetate (PMA; Sigma-Aldrich) and ionomycin calcium salt (Sigma-Aldrich) to determine T-cell reactivity to non-specific stimuli. Viable cell counts were determined using trypan blue dye exclusion on a haemocytometer (Sigma-Aldrich). From each TIL culture, 5 × 10^4^ cells were plated in each well of a 24-well plate and a total of 4 wells were plated for each sample: one negative control and one positive control each for 24, 48, and 72 h TILs in the negative control wells were seeded with CM; TILs in the positive control wells were seeded in 1 mL of stimulating media (SM; Table 4) [39]. The 24-well plate was then incubated at 37 °C with 5% CO_2_ and the supernatants in each well were collected at the respective time points. Supernatants were stored in –80 °C until further use. The collected supernatant samples were tested for IFNγ levels using a commercial IFNγ ELISA kit (Thermo Fisher Scientific). A subset of samples was also tested for granzyme B secretion using a commercial granzyme B ELISA kit (Invitrogen, Waltham, MA, USA). Absorbance readings and data analysis were completed with a SoftMax Pro 5 (Molecular Devices, San Jose, CA, USA).

## 3. Results

### 3.1. Tissue Specimens

A total of 92 tumour specimens were cultured using the TFM outlined in the methods section. Patient characteristics are listed in Table 1. TIL34 and TIL69 were specimens from the same patient; TIL74 and TIL146 were specimens from the same patient; TIL161 and TIL175 were specimens from the same patient; TIL64 and TIL190 were specimens from the same patient. Most specimens were superficial lesions (*n* = 65), some were deep lesions (*n* = 12), and some were bone lesions (*n* = 15). Most specimens were primary tumours (*n* = 81), some were metastatic tumours (*n* = 11). Forty-seven specimens were from male patients and 40 specimens were from female patients. The average age at the time of resection was 59 years (±18.5 years; median 62 years). Most patients had high grade tumours (*n* = 58) and some were systemic relapses from previous lesions (*n* = 32). All but 10 patients had no prior treatment. One patient had pre-operative radiation on the primary tumour; three patients had prior radiation on the primary tumour at least 1 year earlier; and six patients had pre-operative chemotherapy. The specimens were subtyped based on histology and spanned across five sarcoma subtypes: MFS, UPS, OS, liposarcoma (LPS), and leiomeiosarcoma (LMS). There were similar sample sizes across the five sarcoma subtypes.

### 3.2. TIL Expansion from Sarcoma Specimens

Wells with similar cell densities were pooled together and an initial cell count was performed at week 1. The average proportion of “high-density” wells ranged from 13% from LMS specimens to 44% from MFS specimens. The average proportion of “low-density” wells ranged from 26% from OS specimens to 43% from LMS specimens. The average proportion of “no growth” wells ranged from 19% from MFS specimens to 51% from OS specimens (Figure 2).

Overall, there were 6 “high-density” populations that yielded >10^7^ cells (Figure 3a), which is a comparable count to those previously published [35]. Around 50% of the “low-density” populations had <10^5^ cells (Figure 3b), while only 30% of the “high-density” populations had a cell count of less than 10^5^ cells.

After a 3-week expansion, the initially “high-density” population overall had significantly more cells than the “low-density” populations, but there were no significant differences in the growth rate (data not shown). This suggests the difference in cell density at week 1 may be attributed to an uneven distribution of TILs within the original tumour specimen (e.g., heterogeneity, possibly attributable to intratumoural lymphoid aggregates). Wells that were assigned as “low density” may have started out with fewer TILs which resulted in the lower cell density. At week 1, this difference in cell counts between the “high-density” and “low-density” populations was only significant overall, within the MFS subtype, and UPS subtype; however, at week 3, within the MFS and UPS subtype, the difference became insignificant, indicating a smaller difference between the “high-density” and “low-density” populations over time. This smaller difference in cell count overtime could suggest TIL compositions with different proliferative capacity between the “high” and “low” density populations. While there were no significant differences in growth rates, the decreasing difference in cell count suggests that cells in “low-density” populations could have comparable proliferative capacity to cells in “high-density” populations.

#### In Vitro Expansion

The cell count varied from 2.5 × 10^3^–5.8 × 10^7^ cells across the “high-density” populations, and 0–6.5 × 10^6^ cells across the “low-density” populations. The visible difference in cell density at week 1 was not an accurate indicator of in vitro cell growth as a “high-density” population from one tumour specimen may have less cell yield than a “low-density” population from another. In order to better, and more objectively, evaluate a patient’s “potential” for TIL therapy a “successful” tumour expansion was defined as having >1.0 × 10^5^ cells at week 3 in either the “high-density” or “low-density” population. This threshold was determined in consideration of the required cell counts needed to conduct subsequent experiments, and also the minimum cell count that would yield a clinically significant number of cells using a rapid expansion protocol described in Nguyen et al. [35]. The definition of a “successful” expansion was set at 1 × 10^5^ cells at week 3 based on the requirements for the rapid expansion protocol, which induces a 500- to 2000-fold expansion and is typically needed to obtain the large number of cells (>10^10^) used for ACT transfusion [35]. Most studies have reported using an initial seeding density of 0.5–1 × 10^6^ cells; however, the TIL yield from sarcoma specimens during the initial expansion phase is considerably lower in comparison to other cancer types, such that the benchmark of success was adjusted to reflect that difference [23,35].

Across 92 tumour specimens, TILs were successfully expanded from 54 tumour specimens, with only 11 tumour specimens yielding no TILs at the end of week 1. The overall success rate of expansion from sarcoma tumour specimen was 59.1%. MFS (*n* = 25; 80%) had the highest success rate in TIL expansion in comparison to UPS (*n* = 22; 50%), OS (*n* = 17; 52.9%), LPS (*n* = 17; 47.1%), and LMS (*n* = 11; 54.5%) (Figure 4).

### 3.3. TIL Expansion Success Not Correlated with Either Overall or Disease-Free Survival

Survival analysis was performed for patients with a first presentation of sarcoma and localized disease. Overall and disease-free survival for this cohort at two years post-surgical resection was 67.9% and 53.4%, respectively. There was no association with TIL expansion success and either overall or disease-free survival (data not shown). Overall survival for the “fail”, “no wells” and “success” groups at two years was 81.5%, 38.1% and 68.2%, respectively (*p* = 0.43). Disease-free survival for the same groups at two years was 60.2%, 50.8% and 51.2%, respectively (*p* = 0.87). Further analysis was done and adjusted to include a “success” group and a combined group of the “fail” and “no wells” groups. Overall survival for the ‘fail/no wells’ groups at two years was 68.4% and 68.2%, respectively (*p* = 0.54). Disease-free survival for the same group at two years was 56.8% and 51.2%, respectively (*p* = 0.62).

### 3.4. Phenotype of Sarcoma TILs

Morphologically, healthy TIL cultures displayed distinct clusters of proliferating cells (Figure 1). Flow cytometry was conducted to identify distinct immune cell types. As the minimum cell count required for flow cytometry analysis is 1.5–2.0 × 10^6^ cells, in the subsequent studies, all characterized populations were derived from tumour specimens that had successful expansions. Preliminary characterization of 38 TIL populations derived from 25 tissue specimens (25 “high-density” populations and 13 “low-density” populations) were performed using six markers: CD3 (T-cells), CD14 (monocytes), CD19 (B-cells), CD56 (NK cells), CD4 (helper T-cells), and CD8 (cytotoxic T-cells).

#### Difference in Growth Density Driven by Unequal Distribution of TILs

As reported by Crome et al., CD3^–^CD56^+^ cells are a distinct population that suppresses TIL expansion and cytokine production [40]. When comparing the “high-density” and “low-density” populations derived from the same tumour specimen (*n* = 13), there were no significant differences in the proportion of CD3^–^CD56^+^ cells (Figure 5).

There was no significant difference in the proportion of CD3^–^CD56^+^ cells in expanded TIL cultures and the blood control (*p* = 0.31), which indicates that the lower cell yield from sarcoma specimens is not attributed to CD3^–^CD56^+^ cells. There were also similar proportions of CD3^–^CD56^+^ cells in the “high-density” and “low-density” populations derived from the same tumour specimen, indicating that the visible difference in growth density under the microscope at week 1 was not likely driven by CD3^–^CD56^+^ cells, but rather, uneven distribution of TILs in the primary tumour specimen [23]. The lower TIL yield from “low-density” populations may be due to limited availability of TILs rather than a difference in replicative potential.

The proportions of CD3^+^CD4^+^ and CD3^+^CD8^+^ cells varied between each tumour specimen and also showed variation within certain patients where 2 tumour specimen collected at different timepoints were cultured. The variation in vitro between tumour specimens could be due to different levels of TIL infiltration. The variation in vitro within patients could be due to uneven distribution of TILs within the tumour; hence the tumour specimen used for each biological replicate may also have different levels of infiltration.

### 3.5. Abnormal CD4/CD8 Ratios in Sarcoma TIL Cultures

All TIL populations contained CD3^+^, CD4^+^, and CD8^+^ cells (Figure 6). The proportions of CD3^+^ cells in TIL populations were not significantly different from the blood control (*p* = 0.09). In “high-density” populations, the proportion of CD4^+^ cells ranged from 1–99% and CD8^+^ cells ranged from 0.2–99%. In “low-density” populations, the proportion of CD4^+^ cells ranged from 11–99% and CD8^+^ cells ranged from 0.1–87%. The proportions of CD3^+^CD4^+^ and CD3^+^CD8^+^ cells varied between each tumour specimen and also showed variation within certain patients, where 2 tumour specimen, collected at different timepoints, were cultured. The variation in vitro between tumour specimens could be due to different levels of TIL infiltration. The variation in vitro within patients could be due to uneven distribution of TILs within the tumour; hence the tumour specimen used for each biological replicate may also have different levels of infiltration. 

The normal CD4/CD8 ratio in a healthy adult range from 1.5–2.5, depending on age, sex, ethnicity, and genetics [41,42]. The CD4/CD8 ratio of sarcoma TILs ranged from 0.01–550 in “high-density” populations, and 0.1–927 in “low-density” populations (Table 5). There was also no strong correlation between the PD-L1 expression in bulk tumour and the proportion of CD3^+^, CD4^+^ and CD8^+^ cells in post-expansion TIL cultures.

Given that the ideal TIL population to expand is tumour-specific CD8^+^ T-cells, it may be better for TIL cultures to have a low CD4/CD8 ratio. A lower ratio could be indicative of higher infiltration of tumour-specific cytotoxic CD8^+^ T-cells or perhaps a TIL culturing method that selectively expands CD8^+^ T-cells.

It is clear that there are abnormal ratios of CD4^+^ and CD8^+^ cells within the tumour microenvironment. Although these ratios were calculated based on post-expansion values, the expansion period was kept to a minimum to better reflect the native population in the tumour specimens. This is also assuming that CD4^+^ and CD8^+^ T-cells were able to replicate at a similar rate, thereby retaining a ratio that is representative of the native TIL population.

### 3.6. Expanded TIL Populations Exhibit Both Tumour-Promoting and Tumour-Suppressing Phenotypes

Fourteen TIL populations from 14 tumour specimens were further characterized via flow cytometry for the markers: CD69, HLA-DR, T-bet, CD45RA, Ki67, CD39, PD-1, and FOXP3.

#### 3.6.1. Expanded TILs May Display an Activated Phenotype Displayed via CD69 and HLA-DR Expression

Based on CD69 and HLA-DR expression, over 80% of the CD4^+^ and CD8^+^ cell populations were activated (Figure 7 and Figure 8). This suggests that the TILs expanded were primed against an antigen, possibly a specific tumour antigen.

CD69 is regarded as an early activation marker can be detected 2–3 h after the T-cell receptor and CD3 interaction [43]. It has also been described as a metabolic gatekeeper as it regulates the secretion of IFNγ, IL-17 and IL-22 [43]. While the exact role of CD69 expression in immune remains unclear, CD4^+^CD69^+^ T-cells have been associated with immune tolerance and Treg characteristics; CD8^+^CD69^+^ T-cells are characteristic of resident memory T-cells [43,44,45].

Similarly, HLA-DR is a marker of T-cell activation; however, more recent studies have associated HLA-DR expression to immune suppressive CD8^+^ T-regulatory cells (Tregs) [46,47,48,49,50]. These CD8^+^ Tregs mediate the immune response via cell-to-cell contact and were able to suppress proliferation of PBMCs [48].

Given that HLA-DR is a prominent activation marker, the HLA-DR^+^ cells from the expanded TIL populations may not be CD8^+^ Tregs and simply be activated CD8^+^ T-cells. Considering that sarcoma TIL cultures showed successful expansion and proliferation in vitro, it is likely that the subsets of CD4^+^CD69^+^ T-cells and CD8^+^HLA-DR^+^ T-cells did not impose suppressive actions and that the markers may be indicative of an activated T-cell population. Since the IL2 supplemented media is not sufficient to activate naïve T-cells and IL2 receptor expression is restricted to antigen-activated T-cells, the results suggest that these activated T-cells were already specific to an antigen in the tumour microenvironment [51].

#### 3.6.2. CD4^+^ Tregs Characterized in Post-Expansion TIL Populations

CD4^+^FOXP3^+^ Tregs are T-cells with immune suppressive activity that typically mediate peripheral tolerance and prevents autoimmunity [44]. CD4^+^FOXP3^+^ Tregs were characterized in all 14 TIL cultures (Figure 9). The difference between the proportions of FOXP3^+^ and FOXP3^–^ CD3^+^CD4^+^ T-cells ranged from 16% (TIL57) to 85% (TIL82). Naturally occurring Tregs are rare, so the presence of Tregs in the TIL populations may be indicative of Tregs in the tumour environment; however, Tregs can also be induced in vitro with Treg-inducing factors, which can include IL2 [52,53]. While the source of the Tregs is unclear, seeing that they are a tumour-promoting population in the tumour microenvironment, it would be desirable to depleted from the TIL population.

#### 3.6.3. Functionality and Life Span of TILs May Be Reduced Due to In Vitro Expansion

There are two important factors to consider when TILs are used for ACT: their tumour specificity and their killing capacity. Based on the expression of activation markers (CD69 and HLA-DR), the expanded TILs likely exhibit an activated state and suggest certain degree of tumour specificity. On the other hand, the killing capacity of TILs can be inferred from the expression of differentiation markers.

The TIL populations were stained for T-bet, an essential mediator of T-cell differentiation and function, and CD45RA, a common marker of naive T-cells [54,55,56]. T-bet expression is crucial for T_H_1 differentiation and IFNγ production, which are important for orchestrating tumour-specific immune responses [54,55,57,58]. CD45RA expression has also been associated with terminally differentiated memory T-cells, which are reported to have low proliferative capacity and high sensitivity to apoptosis [59,60]. Across the 14 TIL cultures that underwent further characterization, the analysis revealed an average of 39% ± 0.21% CD4^+^T-bet^+^ (gated in CD4^+^) T-cells (median 34%) and 47% ± 0.21% CD8^+^T-bet^+^ (gated in CD8^+^) T-cells (median 48%) (Figure 10). There were low proportions of CD45RA^+^ T-cells with an average of 1.5% ± 1.8% CD4^+^CD45RA^+^ (gated in CD4^+^) T-cells (median 0.94%) and 9.7% ± 11% CD8^+^CD45RA^+^ (gated in CD8^+^) T-cell (median 8.6%) (Figure 11). The moderate T-bet expression suggests that TILs are mature and the low CD45RA^+^ expression suggests that the TILs demonstrate a “non-naive” phenotype. In conjunction, these results indicate an active and differentiated population. Considering that T-cell differentiation occurs in the thymus, these results further suggest that the expanded TILs are activated and specific to an antigen in the tumour microenvironment

Further characterization also revealed expressions of PD-1 and CD39 within the TIL populations, which can be indicative of a tumour-promoting environment. Across the 14 TIL populations, there were on average 43% ± 0.30% CD4^+^PD-1^+^ (gated in CD4^+^) T-cells (median 37%) and 42% ± 0.20% CD8^+^PD-1^+^ (gated in CD8^+^) T-cells (median 45%) (Figure 12). The moderate proportions of PD-1^+^ T-cells suggests that the TILs have adapted an immune suppressing mechanism upon exposure to the tumour microenvironment or from the expansion process.

There was on average 61% ± 28% CD4^+^CD39^+^ (gated in CD4^+^) T-cells (median 65%) and 61% ± 27% CD8^+^CD39^+^ (gated in CD8^+^) T-cells (median 67%) (Figure 13).

In CD8^+^ T-cells, CD39 expression is associated with reduced production of immune promoting cytokines and increased expression of co-inhibitory receptors [61]. The presence of exhausted CD8^+^ T-cells can be detrimental as they provide poor immune-mediated control of tumours and can acquire a regulatory phenotype that is tumour promoting [62,63]. Similarly, CD4^+^CD39^+^ T-cells are metabolically stressed and have increased susceptibility to apoptosis [64]. Typically, increased CD39 expression has been associated with increasing age, thereby contributing to age-associated immune regression. In a tumour-immune context, CD39 expression on activated CD4^+^ T-cells is indicative of exhausted T-cells that carry out effector functions, where CD39 contributes to T-cell attrition and clonal contraction of CD4^+^ T-cells [64]. The high proportion of CD39^+^ T-cells in the TIL population suggests that the T-cells are highly exhausted and may have deteriorated effector functions.

Additionally, the cells were stained for Ki67 as a measure of proliferative capacity. The proportion of Ki67^+^ T-cells was low which suggests that the TILs are susceptible to apoptosis as Ki67 is widely associated with proliferative capability [65,66]. Overall, Ki67^+^ T-cells made up less than 30% of the entire CD3^+^ population (Figure 14).

These results suggest that most of the expanded TIL populations have low proliferative capacity at week 3. Despite being cultured in media conditioned for TIL expansion, it seems that the 3 week expansion period was an appropriate time range for maximizing TIL yield. Given that the REP requires an extended expansion period, there may be a greater proportion of exhausted TILs and also the possibility of an inadequate TIL yield; however, with the help of feeder cells and selective antibodies, these exhausted phenotypes may be alleviated [26].

Alternatively, it has also been suggested that Ki67 expression is associated with antigen-specific T-cell proliferation rather than the innate proliferative capacity of T-cells [56]. This may explain the low proportion of Ki67^+^ T-cells within the TIL populations since they were cultured in CM without any known antigens present.

### 3.7. Correlation of Specific Cell Populations to Each Other

To investigate the relationship between specific cell populations, the proportions of selected cell populations were compared to each other. When several immune cell markers are co-expressed, there are phenotypes more associated with tumour suppression and some more associated with tumour progression (Figure 15)

T-bet expression has been associated with effector functions in T_H_1 cells and CD8^+^ T-cells [67,68]. More specifically, T-bet facilitates the trafficking of T_H_1 cells to inflammatory sites and represses inhibitory effects in exhausted CD8^+^ T-cells. In both CD4^+^ and CD8^+^ populations, the proportion of T-bet^+^ cells had a strong negative correlation with the proportion of PD-1^+^ cells (CD4^+^: r = −0.66; CD8^+^: r = −0.67), suggesting that there is a TIL population with anti-tumour characteristics (Figure 16).

Within the CD4^+^ population, the high correlation of Ki67^+^ and HLA-DR expression (r = 0.76) suggest that proliferating cells were mostly in late activation based on; however, some proliferating cells may be CD4^+^FOXP3^+^ Tregs as there was a moderately positive correlation between the proportion of CD4^+^FOXP3^+^ Tregs with Ki67^+^ cells (r = 0.54; Figure 16a) and with HLA-DR^+^ cells (r = 0.48; Figure 16a).

Similarly, within the CD8^+^ population, proliferating cells were mostly in late activation based on the moderate correlation between Ki67 and HLA-DR expression (r = 0.61; Figure 16b); however, HLA-DR expression has also been associated as CD8^+^ Treg markers [48].

To better differentiate between tumour-suppressing and -promoting CD8^+^ T-cells, CD8^+^ Tregs were reported to lack other classical activation markers, such as CD69, CD25, but increased expression of checkpoint inhibitory proteins, such as CTLA4, LAG3, and PD-1 [50]. Considering that ~40% of CD8^+^ cells were PD-1^+^ there is a possibility that there are CD8^+^ Tregs; however, PD-1 expression is also typical for activated T-cells, suggesting that the presence of PD-1^+^ T-cells are not indicative of CD8^+^ Tregs [69].

### 3.8. Post-Expansion TILs Are Functional and Respond to General Stimulation

Following characterization and confirmation of the CD3^+^CD4^+^ and CD3^+^CD8^+^ cell populations, 20 TIL populations were subjected to generalized stimulation by ION and PMA to determine their innate reactivity post-expansion. All stimulated TIL populations had previously undergone successful expansions. Nineteen TIL populations were previously categorized as “high-density” populations and one was previously categorized as a “low-density” population (TIL220). TILs were cultured in regular CM without the addition of IL2 as the negative control and TILs were cultured in SM for general stimulation.

Data were collected across three time points (24, 48, and 72 h). All stimulated populations had an increase in IFNγ levels compared to the negative control and the IFNγ levels were also sustained for up to 72 h (Figure 17).

A Pearson correlation analysis was conducted between IFNγ levels and the proportion of specific TIL populations. There were no strong correlations between IFNγ levels and the proportion of specific TIL populations. There was a moderate correlation between IFNγ levels at all three time points with the proportion of CD4^+^T-bet^+^ T-cells (24 h: r = 0.68; 48 h: r = 0.44; 72 h: r = 0.53). There was also a moderate correlation between IFNγ levels at all three time points with the proportion of CD8^+^PD-1^+^ T-cells (24 h: r = 0.53; 48 h: r = 0.53; 72 h: r = 0.48).

This suggests that the TILs are functional and able to sustain cytotoxic activity. The ability to sustain cytotoxic effects is essential for TILs as cancer is a chronic disease, and solid tumours will require continuous and prolonged attack.

IFNγ plays a crucial role in cell-mediated immunity by inducing inflammatory mechanisms against immunogens. The main producers of IFNγ are antigen-specific CD8^+^ T-cells; therefore, the results suggest that the TIL populations are composed of functional effector T-cells that have the machinery to perform anti-tumour roles within the TME [70]. In fact, pre-clinical and clinical studies have proven that an induction of IFNγ is required for the activation of a potent anti-tumour immune response [71].

Although there were variations in IFNγ levels across TIL cultures within the same subtype, overall, IFNγ levels upon general stimulation were similar across TILs derived from MFS, UPS, OS, and LMS tumour specimens. The TIL cultures from LIPO tumour specimens did not yield enough cells for the IFNγ functional assay, thus there were no data points. The variation in IFNγ levels observed between TIL cultures does not appear to be driven by the tumour subtype derived cultures, but perhaps, there are patient-specific factors that may affect the innate cytotoxic capacity of TILs. Alternatively, since the composition of the TIL populations was not controlled or standardized upon stimulation, the variation in IFNγ levels may also result from having varying proportions of CD4^+^ and CD8^+^ T-cells. There were no strong correlations between the CD4/CD8 ratio and IFNγ levels; however, there were mildly negative correlations (r = −0.25–0.18), which suggests that having a greater proportion of CD8^+^ T-cells may enhance the cytotoxic capability of TIL populations.

## 4. Discussion

Between sarcoma subtypes, there were varying success rates in the expansion of TILs, which suggests that certain subtypes may have more potential for ACT using TILs. However, there were varying levels of success across tumour specimens of the same subtype, hence there are likely other factors aside from subtype that could affect a patient’s potential for TIL therapy.

An important obstacle to effective ACT may be the presence of immunosuppressive regulators in the tumour microenvironment. While there are multiple regulators that have been identified, such as tumour-associated macrophages and myeloid-derived suppressor cells, in the context of ACT, a concerning mediator of the tumour microenvironment is CD4^+^ Tregs, which have been characterized as CD4^+^FOXP3^+^CD25^+^ [72]. Originally, the identifying marker for CD4 Tregs was CD25; however, FOXP3 was found to be a more definitive marker of Tregs as CD25 is also expressed by effector T-cells [73].

Across the 14 TIL populations that were further characterized, most had a greater proportion of CD4^+^ T-cells that were FOXP3^–^, indicating a larger proportion of CD4^+^ effector T-cells. Seeing that there were TIL cultures with a greater proportion of CD4^+^FOXP3^+^ T-cells, it is unclear whether the in vitro expansion environment induced FOXP3 expression or there were already greater proportions of CD4^+^FOXP3^+^ T-cells within the tumour specimens prior to expansion.

While CD4^+^ Tregs have been a widely recognized regulator of the tumour immune microenvironment, more recently, CD8^+^ Tregs have also been identified, albeit with less definitive markers. Based on the transcriptional profile of CD8^+^ Tregs, the most common marker was CD25^+^CD28^–^, but other markers include HLA-DR, FOXP3, CD103, CD122, and CTLA4 [50,74,75]. As mentioned previously, the CD8^+^ T-cells from the TIL populations did not show characteristics of CD8^+^ Tregs, suggesting that the CD8^+^ population are mostly effector T-cells.

Future studies could include additional considerations for minimizing CD4^+^ and CD8^+^Treg expansion to optimize the in vitro environment and maximize TIL yield. Eliminating Tregs may be an essential step to maintaining a tumour-suppressing TIL population in vitro and subsequently in patients who receive ACT.

Despite the limitations and barriers to effective immunotherapy, specifically ACT, the mobile and robust nature of T-cells can still be capitalized. While the effects and efficacy of ACT against solid tumours remain undetermined and need to be further investigated, this study shows that TILs from sarcoma patients were viable and reactive post-expansion. In conjunction with other studies supporting the idea that TILs are reactive to tumour antigens, ACT can potentially be useful to complement conventional treatments for sarcoma as neoadjuvant or adjuvant therapy [23,35]. For example, perhaps ACT can be used to monitor the disease through TILs circulating in the blood, which could in theory detect rogue tumour cells and prevent distant metastasis. Alternatively, ACT can also be used near the completion of a patient’s treatment as a form of less invasive and intensive therapy to eradicate any remaining cells. This may be effective for patients who are responsive to immune checkpoint therapy, as a study in melanoma demonstrated that responding patients have higher CD8^+^ T-cells pre-treatment, and also an increased proliferation of CD8^+^ T-cells following immune checkpoint therapy [76,77].

What is clear is that the current technology of ACT is not yet sufficiently advanced to eliminate whole solid tumours efficiently and effectively, even with the use of TCRs specific to tumour neoantigens. This may be due to the innate nature of T-cells having a physiological role normally in the human body, therefore there is a limit to the amount of damage T-cells can incite. Rather than modifying T-cells to function beyond this limit, perhaps this natural checkpoint can be used to moderate ACT treatment and prevent unnecessary side effects affecting patients.

In comparison to similar studies in other cancers, such as melanoma and non-small cell lung cancer, the number of TILs collected from sarcomas is relatively low; however, this does not lessen the potential for sarcoma patients to utilize ACT as a treatment option. Rather, this indicates that sarcomas, where some subtypes have been described in the literature as relatively ‘cold’ tumours, still have the potential to be treated by ACT, and more specifically by TILs. PD-L1 expression did not correlate with TIL yield, therefore, future studies can focus on identifying markers that correlate with in vitro TIL expansion and further optimizing the expansion protocol by incorporating steps for rapid expansion. Improvements in the rapid expansion protocol have been reported to expand TIL counts by 500- to 9000-fold [35,39] and studies using cytokines may improve sarcoma-specific REP. A more appropriate indicator of suitable patients for ACT is still yet to be identified, and the CD4/CD8 ratio of the primary tumour may be a potential marker. The results from this study have shown varying CD4/CD8 ratios across the TIL cultures and it would be interesting to see if the ratio in the primary tumour was conserved throughout the expansion process. Additionally, the expression of a group of T_H_1 genes related to high endothelial venules/tertiary lymphoid structures has been associated with outcome in some sarcoma subtypes [22]. Thus, certain gene expressions or the presence of tertiary lymphoid structures may be a potential marker to investigate. Furthermore, studies exploring the chemokine profile of the tumour microenvironment are currently ongoing.

Through this study, TILs were successfully expanded from tumours of five sarcoma subtypes with varying levels of PD-L1, which has traditionally been a marker used to distinguish ‘hot’ vs. ‘cold’ tumours. Several studies in sarcoma have shown that some sarcoma subtypes respond to checkpoint inhibitors, albeit at different levels, and in particular, in patients with inflamed or ‘hot’ tumours. While immune checkpoint therapy has had limited success in patients with non-inflamed or ‘cold’ tumours, it is unclear whether the lack of immune infiltration is due to these patients’ inability to develop immunity, or due to other structural factors that prevent immune infiltration, and ultimately priming against tumour neoantigens.

While the TILs in this study have been shown to be reactive towards general stimulation, further investigation on tumour-specific reactivity will need to be conducted to determine their killing capacity towards tumour cells. Future studies can focus on adapting a rapid expansion protocol for TILs to alleviate the limitations identified in this study. This will allow greater capacity for replicates and subsequent experiments, such as monitoring the change in TIL population in the presence of tumour cells, or in the presence of a therapeutic agent. However, to conduct subsequent experiments, TIL expansion will need to be optimized such that increasing the TIL yield does not significantly induce pro-tumour phenotypes.

## 5. Conclusions

Functional TILs can be isolated and expanded from some sarcoma subtypes: MFS, UPS, OS, LPS, and LMS; however, a benchmark of an effective and clinically significant TIL yield will need to be better defined going forward. There is a potential application of TILs in the treatment of sarcoma. Future studies will benefit from evaluating the killing capacity and tumour specificity of isolated TILs.

## Figures and Tables

**Figure 1 cancers-14-00548-f001:**
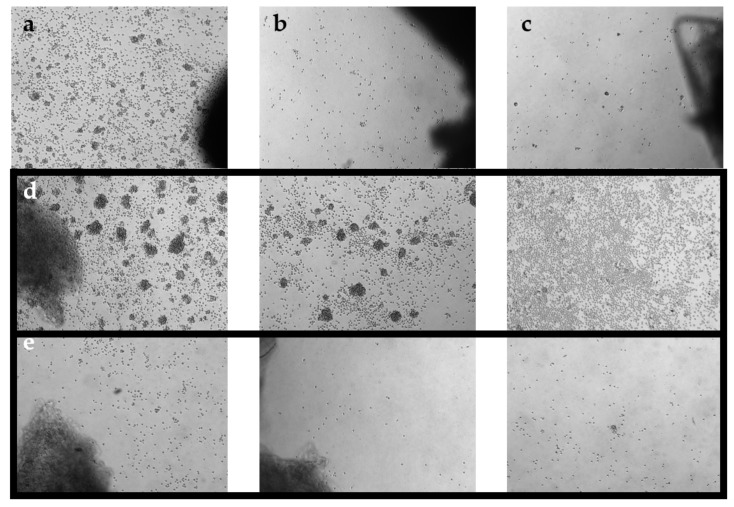
Phase-contrast images of TIL cultures from optimized pooling protocol. Images were taken on day 7 at 5× magnification. (**a**) TIL 36 culture assigned as ‘high density’. (**b**) TIL 36 culture assigned as ‘low density’. (**c**) TIL 36 culture assigned as ‘no growth’. (**d**) TIL 85 cultures assigned as ‘high density’. (**e**) TIL 85 cultures assigned as ‘low density’.

**Figure 2 cancers-14-00548-f002:**
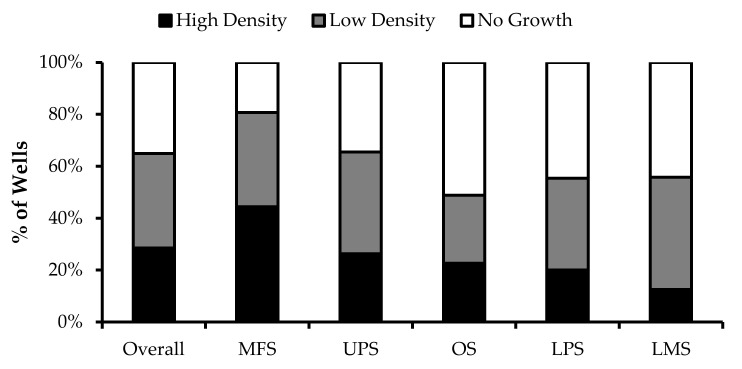
TIL growth variation between wells. Cells were expanded from primary tumour specimens using the tumour fragment method. The average proportion of wells at week 1 that displayed high cell density (distinct TIL clusters), low cell density (scattered TIL growth), and no growth are shown for myxofibrosarcoma (MFS; *n* = 23), undifferentiatied pleomorphic sarcoma (UPS; *n* = 23), osteosarcoma (OS; *n* = 17), liposarcoma (LPS; *n* = 15), and leiomyosarcoma (LMS; *n* = 10).

**Figure 3 cancers-14-00548-f003:**
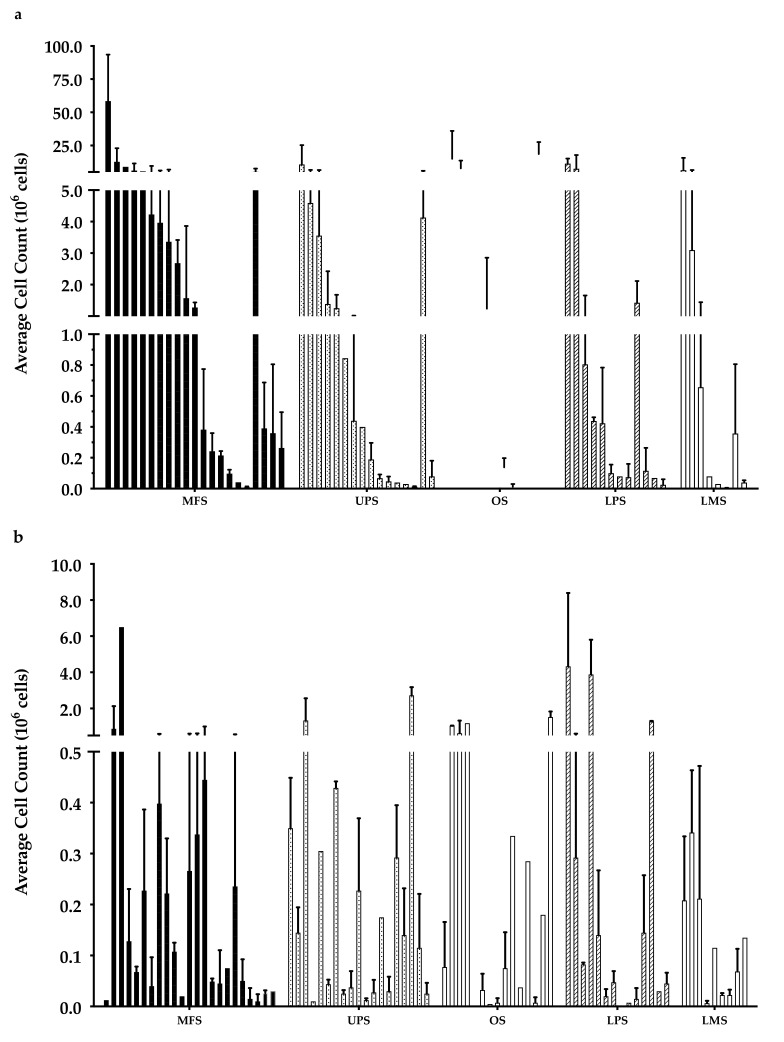
Week 3 average cell count. The graph represents the week 3 cell count of (**a**) “high-density” populations from 21 myxofibrosarcoma (MFS), 16 undifferentiated pleomorphic sarcoma (UPS), 12 osteosarcoma (OS), 12 liposarcoma (LPS), and 8 leiomyosarcoma (LMS); (**b**) “low-density” populations from 23 MFS, 19 UPS, 15 OS, 14 LPS, and 9 LMS). Error bars represent the standard deviation, so bars with error bars indicate the average cell count of biological replicates. Bars without error bars indicate the cell count of one single replicate.

**Figure 4 cancers-14-00548-f004:**
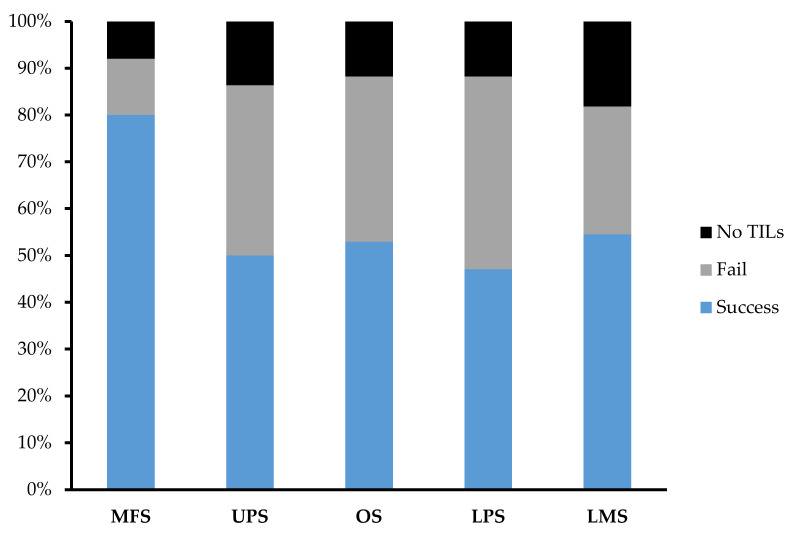
Expansion success rate of myxofibrosarcoma (MFS), undifferentiated pleomorphic sarcoma (UPS), osteosarcoma (OS), liposarcoma (LPS), and leiomyosarcoma (LMS). Graph represents the proportion of tumour specimens with a successful expansion. A successful expansion is defined by having a week 3 cell count that is >1 × 10^5^ cells in either the “high-density” or “low-density” pool. A failed expansion is defined by having a week 3 cell count that is <1 × 10^5^ cells in both the “high-density” or “low-density” pool. No TILs refer to tumour specimens that did not display TIL growth at week 1 and so expansion was terminated after week 1.

**Figure 5 cancers-14-00548-f005:**
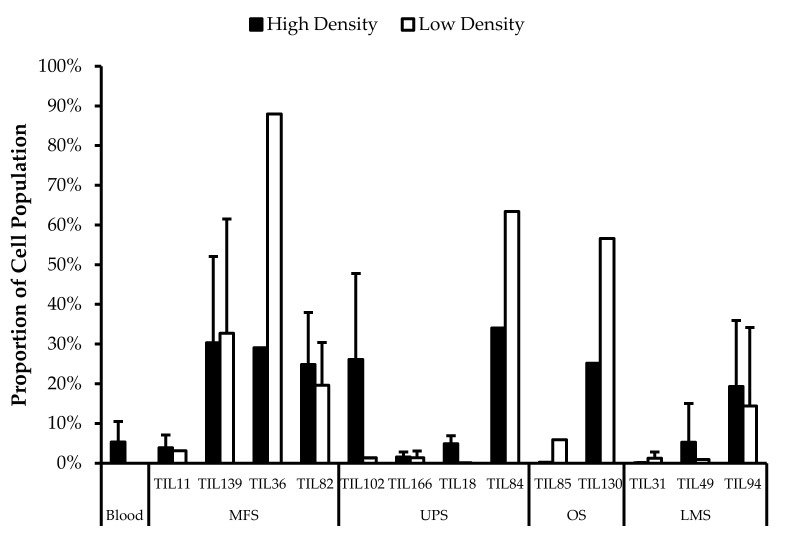
Comparing CD3^−^CD56^+^ cell proportions between “high-density” and “low-density” populations. The graphs represent the TIL composition of “high-density” populations and “low-density” populations (*n* = 13) of thirteen tumour specimens (4 myxofibrosarcoma (MFS), 4 undifferentiated pleomorphic sarcoma (UPS), 2 steosarcoma (OS), and 3 leiomyosarcoma (LMS). Error bars represent the standard deviation, so bars with error bars indicate the average cell count of biological replicates. Bars without error bars indicate the cell count of one single replicate.

**Figure 6 cancers-14-00548-f006:**
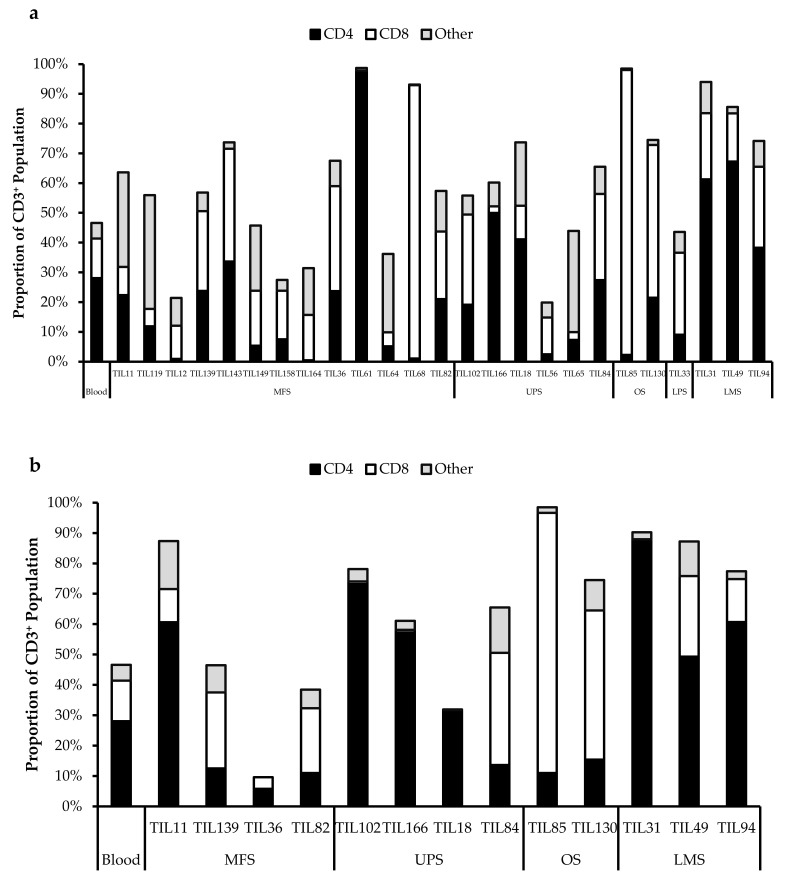
Proportions CD4^+^ and CD8^+^ cells from myxofibrosarcoma (MFS), undifferentiated pleomorphic sarcoma (UPS), osteosarcoma (OS), liposarcoma (LPS), and leiomyosarcoma (LMS) tumour specimens. The graphs represent the TIL composition relative to the CD3^+^ population in (**a**) “high-density” populations (*n* = 25) from 25 tumour specimens and (**b**) “low-density” populations (*n* = 13) from thirteen tumour specimens. Error bars represent the standard deviation, so bars with error bars indicate the average cell count of biological replicates. Bars without error bars indicate the cell count of one single replicate.

**Figure 7 cancers-14-00548-f007:**
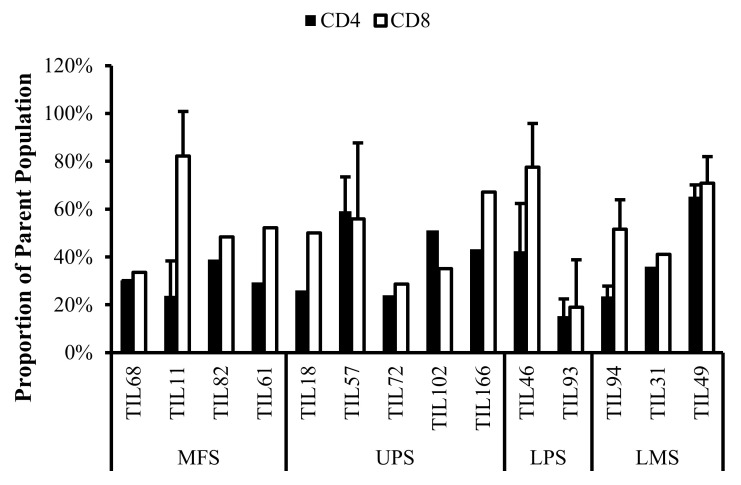
Normalized proportions of CD69^+^ cells in expanded populations from 4 myxofibrosarcoma (MFS), 5 undifferentiated pleomorphic sarcoma (UPS), 2 liposarcoma (LPS), and 3 leiomyosarcoma (LMS) tumour specimens. The proportions of CD4^+^CD69^+^ (gated in CD4^+^) and CD8^+^CD69^+^ (gated in CD8^+^) T-cells are shown for 14 successfully expanded TIL cultures. Error bars represent the standard deviation.

**Figure 8 cancers-14-00548-f008:**
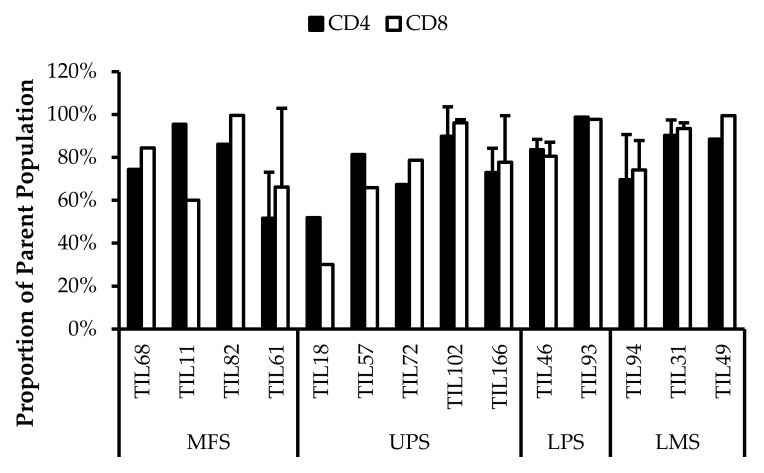
Normalized proportions of HLA-DR^+^ cells in expanded populations from 4 myxofibrosarcoma (MFS), 5 undifferentiated pleomorphic sarcoma (UPS), 2 liposarcoma (LPS), and 3 leiomyosarcoma (LMS) tumour specimens. The proportions of CD4^+^HLA-DR^+^ (gated in CD4^+^) and CD8^+^HLA-DR^+^ (gated in CD8^+^) T-cells are shown for 14 successfully expanded TIL cultures. Error bars represent the standard deviation.

**Figure 9 cancers-14-00548-f009:**
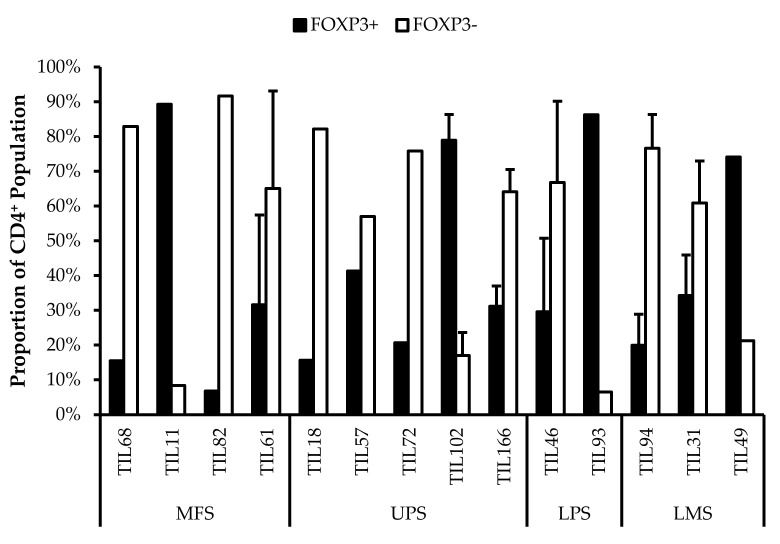
Proportions of FOXP3^+^ cells in expanded populations from 4 myxofibrosarcoma (MFS), 5 undifferentiated pleomorphic sarcoma (UPS), 2 liposarcoma (LPS), and 3 leiomyosarcoma (LMS) tumour specimens. The proportions of FOXP3^+^ and FOXP3^–^ (gated in CD4^+^) T-cells are shown for 14 successfully expanded TIL cultures. Error bars represent the standard deviation, so bars with error bars indicate the average cell count of biological replicates. Bars without error bars indicate the proportion of one single replicate.

**Figure 10 cancers-14-00548-f010:**
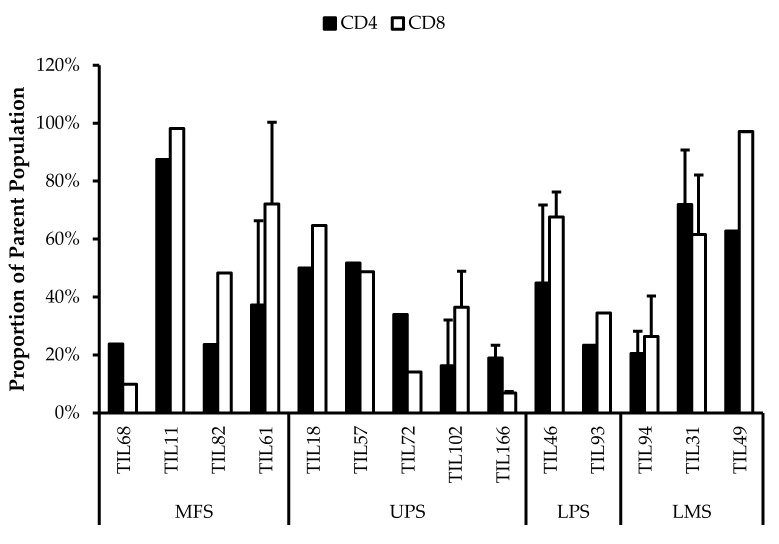
Normalized proportions of T-bet^+^ cells in expanded populations from 4 myxofibrosarcoma (MFS), 5 undifferentiated pleomorphic sarcoma (UPS), 2 liposarcoma (LPS), and 3 leiomyosarcoma (LMS) tumour specimens. The proportions of CD4^+^T-bet^+^ (gated in CD4^+^) and CD8^+^T-bet^+^ (gated in CD8^+^) T-cells are shown for 14 successfully expanded TIL cultures. Error bars represent the standard deviation.

**Figure 11 cancers-14-00548-f011:**
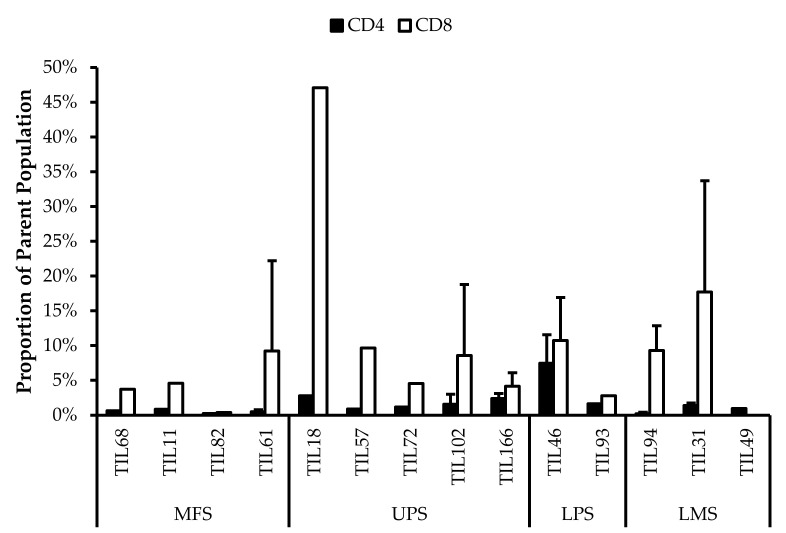
Normalized proportions of CD45RA^+^ cells in expanded populations from 4 myxofibrosarcoma (MFS), 5 undifferentiated pleomorphic sarcoma (UPS), 2 liposarcoma (LPS), and 3 leiomyosarcoma (LMS) tumour specimens. The proportions of CD4^+^CD45RA^+^ (gated in CD4^+^) and CD8^+^CD45RA^+^ (gated in CD8^+^) T-cells are shown for 14 successfully expanded TIL cultures. Error bars represent the standard deviation.

**Figure 12 cancers-14-00548-f012:**
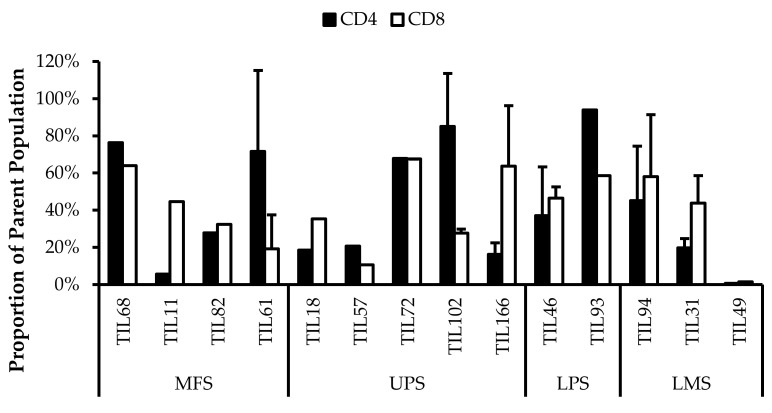
Normalized proportions of PD-1^+^ cells in expanded populations from from 4 myxofibrosarcoma (MFS), 5 undifferentiated pleomorphic sarcoma (UPS), 2 liposarcoma (LPS), and 3 leiomyosarcoma (LMS) tumour specimens. The proportions of CD4^+^PD-1^+^ (gated in CD4^+^) and CD8^+^PD-1^+^ (gated in CD8^+^) T-cells are shown for 14 successfully expanded TIL cultures. Error bars represent the standard deviation.

**Figure 13 cancers-14-00548-f013:**
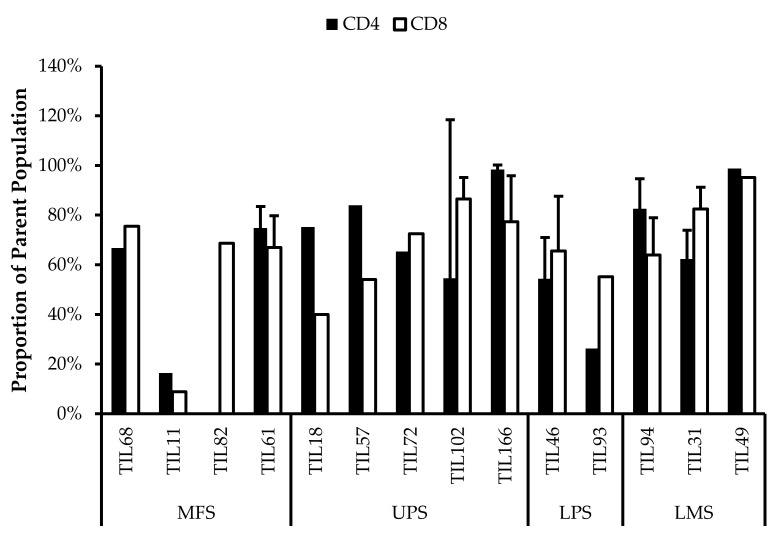
Normalized proportions of CD39^+^ cells in expanded populations from 4 myxofibrosarcoma (MFS), 5 undifferentiated pleomorphic sarcoma (UPS), 2 liposarcoma (LPS), and 3 leiomyosarcoma (LMS) tumour specimens. The proportions of CD4^+^CD39^+^ (gated in CD4^+^) and CD8^+^CD39^+^ (gated in CD8^+^) T-cells are shown for 14 successfully expanded TIL cultures. Error bars represent the standard deviation.

**Figure 14 cancers-14-00548-f014:**
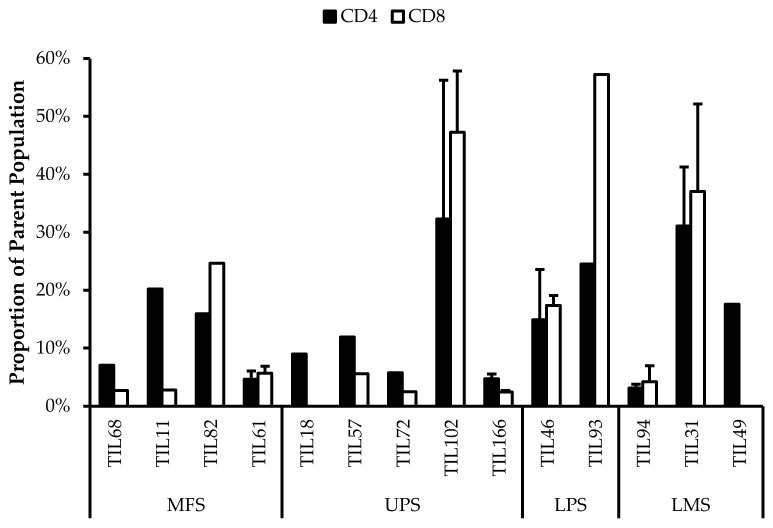
Normalized proportions of Ki67^+^ cells in expanded populations from 4 myxofibrosarcoma (MFS), 5 undifferentiated pleomorphic sarcoma (UPS), 2 liposarcoma (LPS), and 3 leiomyosarcoma (LMS) tumour specimens. The proportions of CD4^+^Ki67^+^ (gated in CD4^+^) and CD8^+^Ki67^+^ (gated in CD8^+^) T-cells are shown for 14 successfully expanded TIL cultures. Error bars represent the standard deviation.

**Figure 15 cancers-14-00548-f015:**
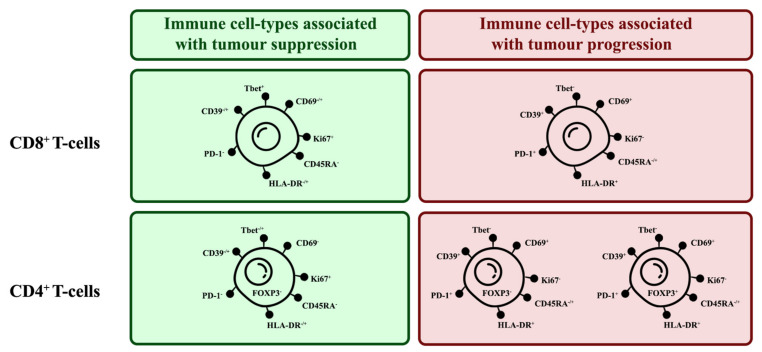
Phenotypes of tumour-suppressing and tumour-promoting TIL subsets. There are key markers expressed by TILs that can indicate their tumour-suppressing or -promoting tendencies.

**Figure 16 cancers-14-00548-f016:**
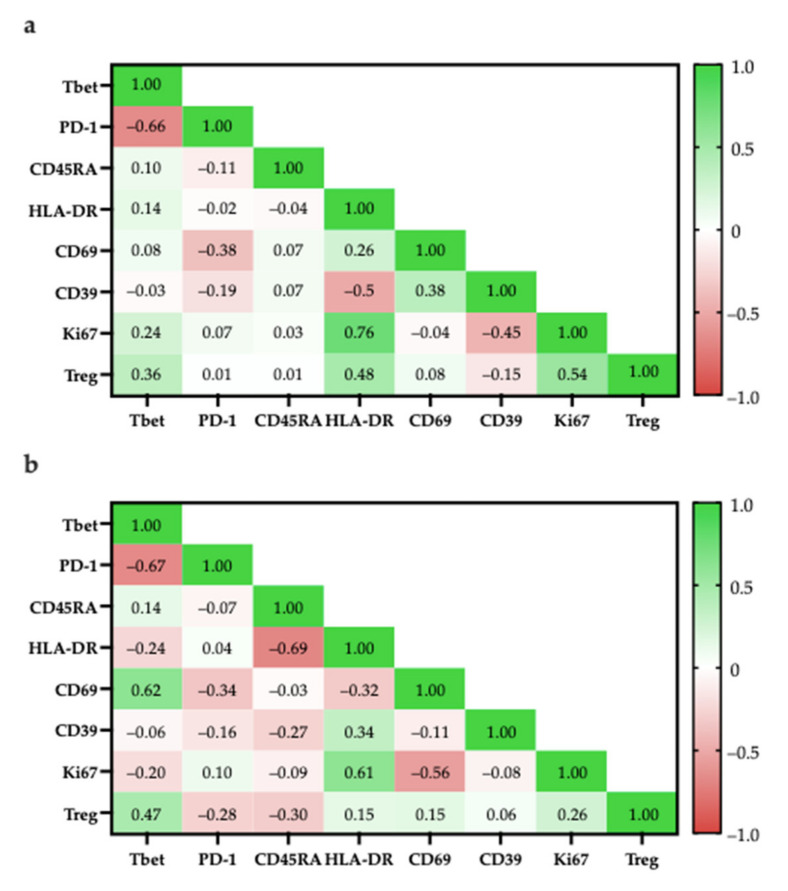
Pearson correlation of specific cell populations. (**a**) Specific cell populations are gated in CD4; (**b**) Specific cell populations are gated in CD8 and Tregs are gated in CD8.

**Figure 17 cancers-14-00548-f017:**
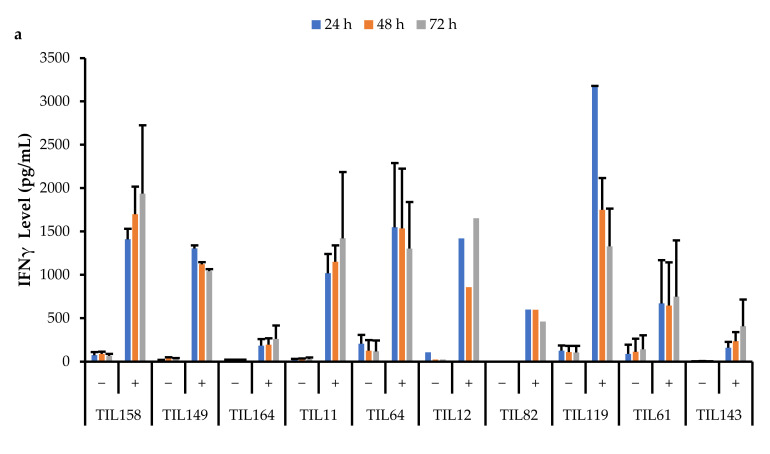
Comparison of IFNγ release by different TIL populations. The graphs represent the average IFNγ levels at 24, 48, and 72 hr for (**a**) 10 myxofibrosarcoma (MFS) TIL populations; (**b**) 5 undifferentiated pleomorphic sarcoma (UPS) TIL populations; (**c**) 2 osteosarcoma (OS) and 3 leiomyosarcoma (LMS) TIL populations. Error bars represent the standard deviation, so bars with error bars indicate the average IFNγ level of biological replicates. Bars without error bars indicate the IFNγ level of one single replicate.

**Table 1 cancers-14-00548-t001:** Demographics and oncologic variables for all patients included in this study.

Variables	Clinical Parameter	No.	%
Number of Patients		87	
Number of Specimen		92	
Mean Age (Range)		59 (18–91)	
Gender	Male	47	54%
	Female	40	46%
Histology	Myxofibrosarcoma (MFS)	25	27%
	Undifferentiated Pleomorphic Sarcoma (UPS)	22	24%
	Osteosarcoma (OS)	17	18%
	Liposarcoma (LPS)	17	18%
	Leiomyosarcoma (LMS)	11	12%
Grade	1	5	5%
	2	34	37%
	3	53	58%
Presenting Status	M0	71	77%
	M1	8	9%
	LR ± Mets	13	14%
Location	Deep	12	13%
	Superficial	65	71%
	Bone	15	16%
Mean Max. Diameter (Range, cm)		12.8 (2.8–39.7)	
Surgical Procedure at Sample Processing	Open Biopsy	38	41%
	Resection	54	59%

**Table 2 cancers-14-00548-t002:** Sequences of the primers used for RT-qPCR.

Primers		Sequences (5′ to 3′)
F ^1^	*STAM2*	TGGATGACAGTGATGCCAATTG
R ^2^	*STAM2*	CGCTGCCTCAGTCTCTATGT
F ^1^	*PD-L1*	TGCCGACTACAAGCGAATTACTG
R ^2^	*PD-L1*	CACTGCTTGTCCAGATGACT

^1^ Forward primer; ^2^ reverse primer.

**Table 3 cancers-14-00548-t003:** Complete media formulation using IMDM.

Material	Final Concentration in CM
Human Serum AB	10%
Penicillin	100 U/mL
Streptomycin	100 µg/mL
Gentamicin sulfate	10 µg/mL
L-glutamine	2 mM
β-mercaptoethanol	5.5 × 10^−5^ M
Human Recombinant IL2	6000 IU/mL

**Table 4 cancers-14-00548-t004:** Stimulating media formulation using IMDM.

Material	Final Concentration in SM
Human Serum AB	10%
Penicillin	100 U/mL
Streptomycin	100 µg/mL
Gentamisin sulfate	10 µg/mL
β-mercaptoethanol	5.5 × 10^−5^ M
PMA	10 ng/ml
ION	500 ng/mL

**Table 5 cancers-14-00548-t005:** Post-expansion CD4/CD8 ratios.

Subtype	Case	CD4/CD8 Ratio	
		High Density	Low Density
**MFS**	TIL11	2.4	5.6
	TIL119	2.0	
	TIL12	0.1	
	TIL139	0.9	0.5
	TIL143	0.9	
	TIL149	0.3	
	TIL158	0.5	
	TIL164	0.03	
	TIL36	0.7	1.5
	TIL61	550	
	TIL64	1.1	
	TIL68	0.01	
	TIL82	0.9	0.5
**UPS**	TIL102	0.6	88
	TIL166	23	66
	TIL18	3.7	225
	TIL56	0.2	
	TIL65	2.8	
	TIL84	0.9	0.4
**OS**	TIL85	0.02	0.1
	TIL130	0.4	0.3
**LPS**	TIL33	0.3	
**LMS**	TIL31	2.8	927
	TIL49	4.1	1.9
	TIL94	1.4	4.3

## Data Availability

Data was contained within the article or supplementary material. The data presented in this study are available upon written request.

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
