# Peer review of "Investigating the Potential of Isolating and Expanding Tumour-Infiltrating Lymphocytes from Adult Sarcoma"

_cancers, 2022, doi:10.3390/cancers14030548_

Round 1

Reviewer 1 Report

Cancers: Investigating the potential of isolating and expanding tumour infiltrating lymphocytes from adult sarcoma (Manuscript ID# cancers-1466766)

This manuscript by Ko et al investigate the potential of isolating and expanding tumour infiltrating lymphocytes from adult sarcoma. Adult sarcomas are notoriously aggressive and difficult to treat, such that searching for new therapeutic opportunities is of high clinical importance. In this manuscript, the authors focus on immunotherapy – specifically, Adoptive Cell Therapy (ACT) – as a new possible approach to sarcoma treatments. The authors leverage against a comprehensive human tumor bank to isolate Tumor Infiltrating Lymphocytes (TILs) from tumor specimens. They then optimize a protocol for TIL expansion and culturing efficacy, successfully culturing TILs from 55 of 92 sarcoma specimens. The authors demonstrate that these approaches could be applicable for 5 subtypes of adult sarcomas.

Overall, a substantive amount of data is presented and well-described. However, the manuscript in its present form reads as if the observations are still a bit preliminary and lack any tangible data that suggest validation of the proposed methods. Thus, the manuscript seems methods/protocol-based without objective pre-clinical connection and is not yet suitable in this reviewer’s opinion for publication in Cancers. A couple of specific comments for the authors to consider:

  • These studies would be significantly enhanced if the authors could correlate any outcome data from their enrolled patients with the TIL results observed.
  • Do the authors have access to previous tissue histologic sections (or could they be generated) that might correlate TIL and/or tumor histology with the TIL results thy observed?

Two formatting issues:

  • Section header “3.1.1 In Vitro Expansion” should be “3.2.1”.
  • The in-text references did not come through correctly upon some aspect of file uploading/conversion.

Reviewer 2 Report

1) The Authors show that TIL's can be cultured with varying degrees of sucess from 5 different sarcoma subtypes. The TIL's cultures seperate into no growth, low density, and high density. Still even the high density cultures are not in sufficient #'s for consideration for ACT therapy, as the authors point out 0.2-2.0 x 1011 are estimated to be needed. How can this be improved. Additionally, the authors attempt to correlate the low and high density cultures to the intrinsic immunogencitiy of the tumors "hot" versus "cold" with PD-1 as the main marker. After PD-1 may not be the best marker of the tumor microenviroment for soft tissue sarcomas. For example teritary lymphoid structures may be a superior marker. Additionlly, recent reports have attempt to further refine up to five subgroups of soft tissue sarcoma with varying amount of immunogencity to the tumor microenviroment. Can the authors clarify the intrinsic immungencity of the tumors that lead to the high density TIL cultures versus the low density populations?

2) multiple references need to be confirmed

Reviewer 3 Report

I think this is a very interesting article highlighting the potential of immunotherapy on sarcoma treatment and deserve to be published. The manuscript is well written, organized and  the references are updated.

Would be interesting to show some pictures of PDL1 in  different sarcomas subtypes if possible.

Round 2

Reviewer 1 Report

The authors are satisfactorily addressed my comments and is suitable for publication if the other reviewers are satisfied.